# Power Spectrum of Acceleration and Angular Velocity Signals as Indicators of Muscle Fatigue during Upper Limb Low-Load Repetitive Tasks

**DOI:** 10.3390/s22208008

**Published:** 2022-10-20

**Authors:** Béatrice Moyen-Sylvestre, Étienne Goubault, Mickaël Begon, Julie N. Côté, Jason Bouffard, Fabien Dal Maso

**Affiliations:** 1Institute of Biomedical Engineering, Université de Montréal, Montreal, QC H3T 1J4, Canada; 2School of Kinesiology and Physical Activity Science, Université de Montréal, Montreal, QC H3T 1J4, Canada; 3Centre de Recherche du CHU Sainte-Justine, Montreal, QC H3T 1C5, Canada; 4Department of Kinesiology and Physical Education, McGill University, Montreal, QC H3A 0G4, Canada; 5Department of Kinesiology, Université Laval, Quebec, QC G1V 0A6, Canada; 6Centre Interdisciplinaire de Recherche sur le Cerveau et l’Apprentissage, Montreal, QC H7N 0A5, Canada

**Keywords:** time-frequency analysis, work task, repetitive pointing task, inertial measurement units

## Abstract

Muscle fatigue is a risk factor for developing musculoskeletal disorders during low-load repetitive tasks. The objective of this study was to assess the effect of muscle fatigue on power spectrum changes of upper limb and trunk acceleration and angular velocity during a repetitive pointing task (RPT) and a work task. Twenty-four participants equipped with 11 inertial measurement units, that include acceleration and gyroscope sensors, performed a tea bag filling work task before and immediately after a fatiguing RPT. During the RPT, the power spectrum of acceleration and angular velocity increased in the movement and in 6–12 Hz frequency bands for sensors positioned on the head, sternum, and pelvis. Alternatively, for the sensor positioned on the hand, the power spectrum of acceleration and angular velocity decreased in the movement frequency band. During the work task, following the performance of the fatiguing RPT, the power spectrum of acceleration and angular velocity increased in the movement frequency band for sensors positioned on the head, sternum, pelvis, and arm. Interestingly, for both the RPT and work task, Cohens’ d effect sizes were systematically larger for results extracted from angular velocity than acceleration. Although fatigue-related changes were task-specific between the RPT and the work task, fatigue systematically increased the power spectrum in the movement frequency band for the head, sternum, pelvis, which highlights the relevance of this indicator for assessing fatigue. Angular velocity may be more efficient to assess fatigue than acceleration. The use of low cost, wearable, and uncalibrated sensors, such as acceleration and gyroscope, in industrial settings is promising to assess muscle fatigue in workers assigned to upper limb repetitive tasks.

## 1. Introduction

Upper limb musculoskeletal disorders (MSDs) are one of the most common types of injuries among industrial workers. They accounted for 30% of declared MSDs in the Quebec province between 1998 and 2007 [1], and for most lost workdays per year in US (Statistics 2015, s. d.). Workers performing low-load highly repetitive tasks, such as poultry processing, hairdressing or dentistry, have more than twice the risk of developing MSDs than those performing various and mobile tasks such as air traffic control, rubber mixing or nursing work [2]. Although muscle activation amplitudes are lower than 20% of their maximum during low-load work activities [2,3], movement repetition causes muscle fatigue [4,5], which has been identified as a risk factor for developing musculoskeletal disorders [6,7,8]. Consequently, the assessment of muscle fatigue in the workplace is critical to improve health prevention in workers performing low-load repetitive tasks [9].

Muscle fatigue alters joint kinematics [10,11]. Indeed, glenohumeral mean and maximum elevation angles decreased during various fatiguing low-load repetitive tasks involving the upper limb [10,12,13,14]. These decreased upper limb ranges of motion were compensated by an increased trunk range of motion during fatiguing sawing [15], reaching [12] and pointing tasks [16]. To date, investigations of kinematic alterations caused by muscle fatigue mostly come from laboratory-based experiments involving optoelectronic or equivalent motion-capture systems [17,18]. Recently, some authors used inertial motion units (IMUs) to assess fatigue-related kinematic alterations [9,19,20]. IMUs have the advantage of being inexpensive, portable, and easy to use when compared to optoelectronic systems [21]. Therefore, IMUs became increasingly popular for monitoring kinematics in clinical and sport research [22,23] and are promising for assessing fatigue directly in the workplace. However, IMUs-based 3D orientation of rigid body segments are calculated using a fusion algorithm combining 3-axis accelerometer, gyroscope, and magnetometer data [24]. In particular, the fusion algorithms integrate gyroscope data, which results in error accumulation over time [25,26,27]. Moreover, the magnetometer is sensitive to environmental magnetic perturbations present in industrial settings [28]. Consequently, when comparing joint angles obtained from IMUs and optoelectronic recordings, the root mean square error on IMUs-based joint angles exceeded 35° for arm flexion and axial rotation during a complex handling task [29]. Furthermore, as orientation errors increase with time [30], it would be impossible to discriminate between fatigue-related and error-related kinematic changes when monitoring work tasks. To compensate for this error accumulation, authors use frequent sensor-to-segment recalibration [12], which is not suitable in the workplace. Interestingly, some studies successfully detected kinematic manifestations of fatigue directly from raw acceleration signal, which is not affected by the aforementioned drifting error. The frequency content of acceleration and angular velocity has been shown to change with fatigue. Indeed, the power spectrum of the acceleration increased in the 2–4 Hz frequency band and above 6 Hz [31,32,33] with the most significant fatigue-induced changes in frequencies above 10 Hz categorized as physiological tremor [34]. Time-frequency analyses of IMU signals may potentiate our ability to detect fatigue during fatiguing low-load repetitive work tasks by concurrently detecting fatigue-related motor alterations which are typically observed in the movement frequency band and in the physiological tremor frequency band.

The objective of this study was to assess the effect of muscle fatigue on the frequency content of acceleration and angular velocity signals during a standardized repetitive pointing task (RPT) known to generate upper limb fatigue and an upper limb low-load repetitive work task (i.e., tea bag filling). For both the RPT and the work task, we hypothesized that the acceleration and angular velocity power spectrum of the trunk would increase in frequency bands around the movement cadences as its range of motion increased with fatigue, which would be the reverse for upper limb segments [35]. We also hypothesized that the power spectrum of upper limb distal segments would increase around 10 Hz with fatigue due to physiological tremor [21,31,36,37].

## 2. Methods

### 2.1. Participants

Twenty-four right-handed participants (12 women; age: 32.9 ± 8.9 years; mass: 66.8 ± 10.9 kg; height: 166 ± 9 cm) were recruited among workers of a tea packaging factory. All participants were familiar with the work task investigated in this study since they had performed this task full-time (23 participants) or part-time (1 participant) for 2.6 ± 1.6 months [range: 1–24 months]. Exclusion criteria included any medically diagnosed upper limb disabilities or musculoskeletal disorders at the time of the experiment. All participants read and signed a written informed consent form before performing any experimental procedure. 

### 2.2. Instrumentation

*Kinematics.* Participants were equipped with 11 IMUs (MTw, Xsens, Enschede, Netherlands; https://www.xsens.com/products/mtw-awinda, accessed on 13 October 2022) positioned on the pelvis, sternum, head and on both hands, lower arms, upper arms, and shoulders. The location of IMUs followed Xsens recommendations [38] except for the shoulders, where they were positioned closer to the acromion to leave space for reflective markers and electromyography electrode positioning (Figure 1A). IMU acceleration and angular velocity were recorded using the MT Manager software (Xsens, Enschede, Netherlands) at a sampling rate of 40 Hz.

*Repetitive pointing task (RPT).* Two cylindrical touch-sensitive sensors (length: 6 cm, radius: 0.5 cm, Quantum Research Group Ltd., Hamble, UK) were used as proximal and distal targets for the RPT (Figure 1B). The latter were placed at shoulder height, in the midsagittal plane, and at 30% (proximal target) and 100% (distal target) of the arm length. Each time the sensors were touched by the participants, TTL pulses were recorded at a sampling rate of 2000 Hz by the Nexus software (Vicon, Oxford, UK) and an auditory feedback was given to participants to help them synchronize with the tempo of the metronome (see Section 2.3).

*Force.* A unidirectional S-shape load cell (363-D3-300-20P3, InterTechnology Inc., Don Mills, ON, Canada) was used to measure the shoulder flexion maximal voluntary isometric force. The load cell was attached to a horizontal bar located above the RPT setup (Figure 1C). Force data were recorded at a sampling rate of 2000 Hz using Nexus (Vicon, Oxford, UK).

IMUs, touch-sensitive sensors, and load cell data were synchronised via a TTL pulse sent by the MT Manager software to Nexus (Vicon, Oxford, UK) at the beginning of each recording.

### 2.3. Experimental Protocol

The experiment was composed of the succession of *maximal voluntary isometric contractions* (MVICs), a *work task*, and a *RPT* as shown on Figure 1E.

*Maximum voluntary isometric contractions*. Participants stood up facing the MVIC setup (Figure 1B) with their shoulder flexed at 90° and their forearm below the horizontal bar attached to the load cell. They were asked to push upward as hard as they could against the horizontal bar for 3 s using shoulder muscles without compensation of the trunk. Verbal encouragement was given to participants. As presented in Figure 1E, MVICs were performed twice at the beginning of the experiment (MVIC_1_ and MVIC_2_), every 2 min during the RPT and at the end of the work task (MVIC_4_). MVIC_3_ corresponds to the last MVIC performed during the RPT (cf *Repetitive pointing task).*

*Repetitive pointing task*. Participants stood upright in front of the RPT setup with the dominant arm held in a horizontal plane at shoulder height, and the feet parallel at shoulder width. They were asked to touch alternatively the two touch-sensitive targets at a rhythm of one cycle (back and forth) per two secs [16,35,39] by synchronising the targets’ sounds to those of an external metronome. To ensure that participants maintained their arm horizontal, an elliptically shaped mesh barrier was placed under the elbow trajectory. The non-dominant arm rested on the side of the body during the whole RPT. Participants reported their rate of perceived exertion every 30 s using the CR-10 Borg scale [40]. Additionally, the RPT was interrupted every 2 min so that participants performed a MVIC as described above. After each MVIC, participants resumed the RPT without resting. Participants were asked to perform the task for as long as possible and were stopped when they reached a score of 8 or higher on the CR-10 Borg scale. Participants were not aware of this stoppage criterion.

*Work task*. The work task was performed before and immediately after the RPT. This task was identified, according to interviews, as being the most repetitive fatiguing task for upper limb muscles, among all work tasks performed at the tea packaging factory. The setup described thereafter was made according to observations made at the workers’ tea packaging factory to replicate actual workplace conditions. Participants performed a repetitive work task consisting of filling tea bags of 57 g using a 35 g shovel in their dominant hand. A scale was positioned on their non-dominant side to adjust the required weight. The table height was 87 cm. Participants were instructed to keep their own personal work pace for a 2 min duration trial. According to the observations made on the workplace, this was expected to correspond to the filling of 6 tea bags.

### 2.4. Data Processing

Data processing and statistical analyses were performed with Matlab R2019a (The MathWorks Inc., Natick, MA, USA). All filters mentioned thereafter are zero-lag second-order Butterworth filters. 

Load cell data were low-pass filtered at 30 Hz. To obtain maximum voluntary isometric force, filtered load cell data were averaged over a 1 s overlapping moving window and the maximum value obtained during each MVIC trial was kept for statistical analysis. 

IMU acceleration and angular velocity data were bandpass filtered between 0.1 Hz and 15 Hz [9,41] and were centered around zero by subtracting their mean value. 

Power spectrum. Power spectrum amplitude. For each sensor of each participant, a time-frequency convolution (Equations (1) and (2)) was applied to the *x*-, *y*-, and *z*-axis acceleration and angular velocity using a continuous Complex Morlet wavelet 8-1 (wave number: 8, frequency range: 0.1 Hz to 15 Hz in 0.05 Hz steps; “cwt” Matlab function).
(1)$$W_{x}\left(s,u\right)=\sum_{k=1}^{T}x\left(t_{k}\right)\bar{\varphi_{s,u}\left(t_{k}\right)}$$
where s and u denotes frequency and time, respectively; $x\left(t_{k}\right)$ is a time series of length T regularly spaced at k; $\bar{z}$ denotes the conjugate of a complex number z.
(2)$$\varphi_{s,u}\left(t_{k}\right)=\frac{1}{\sqrt{s}}\varphi \left(\frac{t_{k}-u}{s}\right)$$

*X*-, *y*-, and *z*-axis time-frequency maps were then summed for acceleration and angular velocity independently. For the RPT, the resulting time-frequency maps were segmented in time based on cycles extracted from touch-sensitive sensor data. Only cycles lasting between 1.6 s and 2.4 s from the first 10 (task Initiation) and last 10 (task Termination) cycles were kept for analysis. Each segment of the time-frequency map was then interpolated over 100 points. Finally, all segments of the time-frequency map were average for task Initiation and Termination, before averaging in the 0.4–0.6 Hz and 6–12 Hz frequency bands. These frequency bands were selected to detect kinematic adaptations to fatigue and tremor related changes, respectively. For the work task, the middle 100 s of the 120 s duration of the work task was kept for analysis, and averaged in the 0.1–4 Hz and 6–12 Hz frequency bands. Note that the lower frequency band was larger for the work task than for the RPT because the main frequency component was larger in the work task, as expected.

### 2.5. Statistical Analyses

*Maximum voluntary isometric contraction.* A one-way ANOVA with repeated measures on Time (MVIC1, MVIC2, MVIC3, MVIC4) was performed on maximum voluntary isometric force, followed by a Tukey post hoc analysis. 

*Repetitive pointing task.* The number of remaining cycles as well as the average and standard deviation of their duration were compared between task *Initiation* and *Termination* using a paired *t*-test to assess the effect of Time on task execution. For each IMU sensor, the 0.4–0.6 Hz and 6–12 Hz power spectrum of acceleration and angular velocity were compared between task *Initiation* and *Termination* using a paired *t*-test for each IMU sensor.

*Work task.* For each IMU, the 0.1–4 Hz and 6–12 Hz power spectrum of acceleration and angular velocity were compared between pre- and post-RPT using a paired *t*-test.

For all analyses, Cohen’s d effect size statistics were reported and qualitatively interpreted as small (d < 0.5), moderate (0.5 < d < 0.8) and large (d > 0.8) according to frequently used standards [42].

## 3. Results

### 3.1. Maximum Voluntary Isometric Force

The one-way ANOVA on maximum voluntary isometric force revealed a significant Time effect (F_3,88_ = 7.130; *p* < 0.001). The post hoc analysis showed that MVIC3 was significantly smaller than MVIC1 (t_23_ = 4.702; *p* < 0.001; Coh. d = 0.70), MVIC2 (t_23_ = 6.157; *p* < 0.001, Coh. d = 0.95), and MVIC4 (t_23_ = −5.926; *p* < 0.001, Coh. d = 1.22) (Figure 2).

### 3.2. Repetitive Pointing Task

*Task execution.* Participants performed the RPT for 306 ± 114 s [range 120–540 s]. As cycles lasting less than 1.60 s and more than 2.40 s were removed, 8.96 ± 2.24 and 9.38 ± 2.06 cycles were included in the analysis for task *Initiation* and *Termination*, respectively. Cycles lasted on average 1.87 ± 0.41 s and 1.88 ± 0.41 s during task *Initiation* and *Termination*, respectively. Their duration variability was 0.10 ± 0.05 s and 0.10 ± 0.04 s for task *Initiation* and *Termination*, respectively. The number of cycles included in the analysis, their mean duration, and variability duration were not significantly different between task *Initiation* and *Termination* (t_23_ = 1.794; *p* = 0.086, t_23_ = 0.523; *p* = 0.606, t_23_ = 0.234; *p* = 0.817, respectively).

*Power spectrum.* The paired t-test performed on power spectrum revealed significant differences for both acceleration and angular velocity between task *Initiation* and *Termination* (Figure 3 and Table 1). Concerning the 0.4–0.6 Hz frequency band, the power spectrum significantly increased for the acceleration and angular velocity of IMUs positioned on the head, sternum, and pelvis and for the angular velocity of the IMU positioned on the shoulder (moderate to large effect size). Alternatively, the power spectrum significantly decreased for the acceleration and angular velocity for the IMU positioned on the hand and for the angular velocity of the IMU positioned on the forearm (small effect size). Concerning the 6–12 Hz frequency band, the power spectrum significantly increased for the acceleration and angular velocity of IMUs positioned on the head, sternum, pelvis, and arm (small to moderate effect size) as well as for the angular velocity of the IMU positioned on the hand (small effect size).

### 3.3. Work Task

The paired t-test performed on power spectrum revealed significant increases for both acceleration and angular velocity between work tasks performed before (pre-) and immediately after (post-) the RPT (Figure 4 and Table 1). Concerning the 0.1–4 Hz frequency band, the power spectrum significantly increased for the acceleration and angular velocity of IMUs positioned on the head, sternum, pelvis, and arm (small effect size). Concerning the 6–12 Hz frequency band, the power spectrum significantly increased for the angular velocity of the IMU positioned on the hand (small effect size).

## 4. Discussion

To the best of our knowledge, the present study is the first to assess manifestations of muscle fatigue using the frequency content of acceleration and angular velocity during low-load dynamic upper limb tasks. During a standardized RPT, the power spectrum of head, sternum, and pelvis acceleration and angular velocity increased between task *Initiation* and *Termination* in all frequency bands of interest. For the 0.4–0.6 Hz frequency band, the power spectrum of the hand decreased. Comparing the pre- and post-RPT tea bag filling work task, we observed an increase in power spectrum of acceleration and angular velocity for the head, sternum, pelvis, and arm. As discussed, these changes in the frequency content of acceleration and angular velocity may be interpreted as indicators of muscle fatigue that could be used in industrial settings to better prevent musculoskeletal disorders.

### 4.1. Repetitive Pointing Task

Increases in effort perception and decreases in maximal voluntary isometric force are characteristics of muscle fatigue [36,43,44]. Firstly, all participants reached 8 or more at the CR-10 Borg scale in 306 ± 114 s. As the task performed by the participants remained the same, the increase of their effort perception is a sign of fatigue [45]. Secondly, the post hoc analysis performed on MVICs data revealed a significant loss of force production immediately after the completion of the RPT confirming that participants developed muscle fatigue during the RPT. Importantly, movement cycle duration and its variability did not change between task *Initiation* and *Termination* suggesting that observed changes in terms of acceleration and angular velocity frequency content can be interpreted as a manifestation of muscle fatigue.

Among these changes, power spectrum in the 0.4–0.6 Hz frequency, which corresponds to the main frequency component of the RPT, increased moderately to largely between task *Initiation* and *Termination* for acceleration and angular velocity for the head, sternum, and pelvis, while it decreased for the hand, which agrees with previous kinematics-based studies. Indeed, postural sway [46], head range of motion and vertical displacement amplitude [47,48], as well as trunk range of motion [15,16,35] have been shown to increase during fatiguing standing and various upper limb tasks such as the RPT used in the present study. Conversely, the decrease of both acceleration and angular velocity power spectrum at the hand may be explained by the decreased elbow range of motion with fatigue as previously reported during RPT [35]. Interestingly, Cohens’ d effect sizes were greater for angular velocity than for acceleration and the angular velocity power spectrum changes also involved the shoulder (increase) and the forearm (decrease), which was not the case for acceleration. These observations suggest that angular velocity may be more sensitive to manifestations of muscle fatigue than acceleration. In addition, for angular velocity, Cohens’ d values were comparable (moderate to high) to those reported from joint range of motion obtained using an optoelectronic system measurements during a RPT [15], suggesting that uncalibrated wearable sensors may be as efficient as complete optoelectronic systems to assess kinematics adaptions to muscle fatigue.

Increases in power spectrum in the 6–12 Hz frequency band, involving the head, sternum, pelvis and arm may be interpreted as physiological tremor known to occur around 10 Hz with muscle fatigue [36,49]. However, to the best of our knowledge, tremor in response to muscle fatigue has been previously mostly observed for distal upper limb segments [31,32,36]. Interestingly, in a study by Kouzaki and Masani [50], a physiological tremor component recorded at the soleus during quiet standing was positively correlated to postural sway measures such as centre of pressure velocity and body acceleration, as well as the 1–10 Hz component of the centre of pressure. Authors suggested that physiological tremor of the soleus muscle may cause the fast components of postural sway during quiet standing. Therefore, it may be suggested that the increased power spectrum of the head and trunk segments results from an increased postural sway during the RPT. Although further studies are needed to better characterise physiological tremor at the head and trunk segments during fatiguing RPT, changes in power spectrum of both 0.4–0.6 Hz and 6–12 Hz frequency bands may be relevant indicators of the manifestation of muscle fatigue.

### 4.2. Work Task

Like the RPT, power spectrum of the 0.1–4 Hz frequency band of acceleration and angular velocity increased for head, sternum, pelvis, and arm segments during the work task. However, the power spectrum also increased for the arm, which was the opposite for the RPT. This discrepancy in terms of power spectrum alteration may be due to task specificities. Indeed, during the RPT that is constrained in terms of timing and targets, the trunk range of motion increases, while it decreases for the upper limb [35]. Alternatively, during the work task, different motor strategy can be employed to fill the tea bags, which may have resulted in different motor adaptations to muscle fatigue that lead to increased power spectrum changes at the arm. Interestingly, Goubault et al. [19] showed that manifestations of muscle fatigue were found at different segments between two different piano tasks involving either restricted range of motion or larger movement amplitudes. These observations emphasize that the location of kinematic adaptations to muscle fatigue measured through the power spectrum changes of acceleration and angular velocity might differ according to the task of interest [36,51]. Nevertheless, in both the RPT and work task, although Cohens’ d effect sizes were smaller for the work task than the RPT, power spectrum of acceleration and angular velocity at the head and trunk segments increased in the frequency band of the movement, reemphasizing the relevance of this indicator for assessing muscle fatigue in the workplaces.

### 4.3. Limitations

Although the RPT successfully caused muscle fatigue, the significant increase in force production between the MVIC3 and MVIC4, performed, respectively, immediately before and after the post-RPT work task, implies some limitations. This increased force production suggests that participants had recovered during the post-RPT work task. Therefore, although our results revealed significant changes in power spectrum of acceleration and angular velocity between pre- and post-RPT measurements, the level of fatigue at the beginning of the post-RPT work task may not have been maintained across the two minutes duration of this work task. Consequently, changes in power spectrum between pre- and post-RPT may not be strictly interpreted as a fatigue effect but rather as changes following the performance of a fatiguing task.

## 5. Conclusions

The objective of this study was to assess changes in the frequency content of acceleration and angular velocity during a fatiguing low-load RPT and the effect of this fatiguing task on an actual tea bag filling work task reproduced in laboratory performed with the upper limb. Power spectrum of acceleration and angular velocity systematically increased with muscle fatigue at the head, sternum, and pelvis in the frequency band of the movement. Alternatively, when significant, the power spectrum changes in distal upper limb were task dependent. The present study shows that the power spectrum of the head and trunk acceleration and angular velocity, recorded using low cost, wearable, and uncalibrated sensors, may be used in industrial settings to assess fatigue in workers assigned to upper limb repetitive tasks.

## Figures and Tables

**Figure 1 sensors-22-08008-f001:**
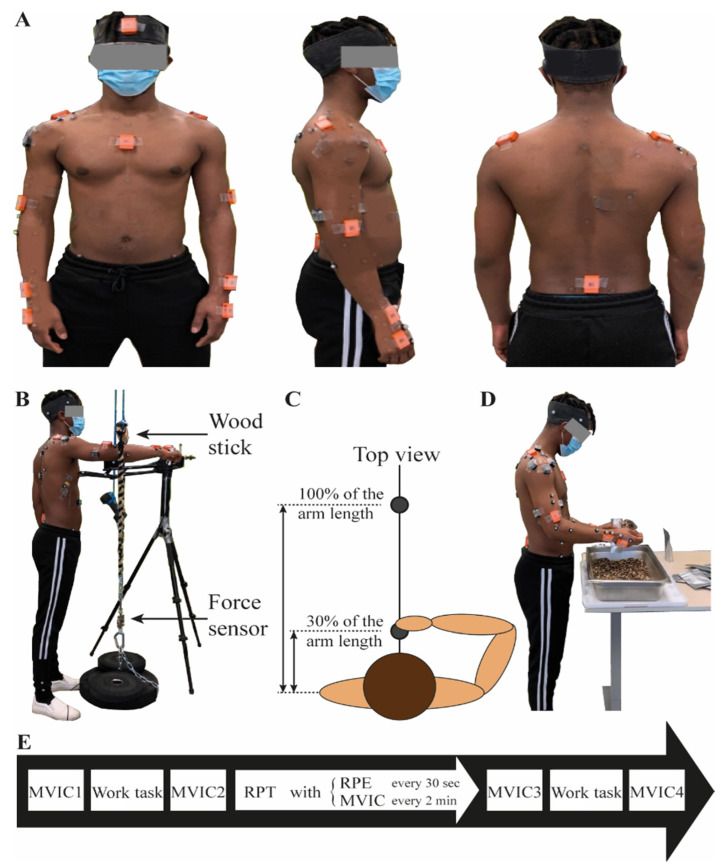
(**A**) Participant equipped with IMUs. (**B**) Picture of the MVIC setup. (**C**) Schematic top view of the RPT. (**D**) Work task. (**E**) Timeline of the experiment.

**Figure 2 sensors-22-08008-f002:**
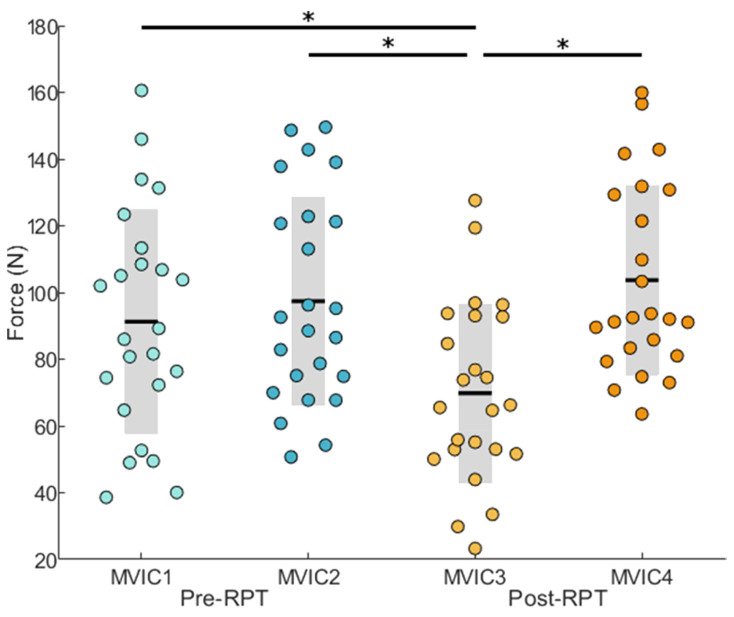
Force developed during MVIC trials. Dots represent each participants’ data. Black horizontal lines represent the participants means; grey bars represent standard deviations. Asterisks represent significant effects at *p* < 0.001.

**Figure 3 sensors-22-08008-f003:**
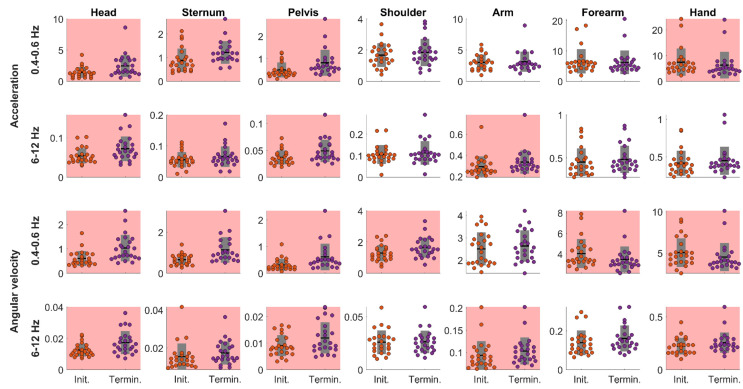
RPT *Initiation* versus *Termination* power spectrum for the acceleration and angular velocity in the 0.4–0.6 Hz and 6–12 Hz frequency bands for the head, sternum, pelvis, shoulder, arm, forearm, and hand. Dots represent each participants’ data. Black horizontal lines represent the participants means; grey bars represent standard deviations. Pink backgrounds indicate significant differences between task *Initiation* and *Termination*.

**Figure 4 sensors-22-08008-f004:**
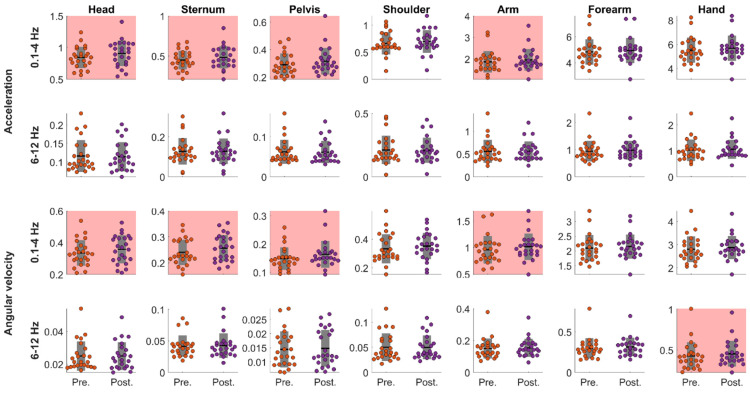
Pre- versus post-RPT work tasks power spectrum for the acceleration and angular velocity in the 0.1–4 Hz and 6–12 Hz frequency bands for the head, sternum, pelvis, shoulder, arm, forearm, and hand. Dots represent each participants’ data. Black horizontal lines represent the participants means; grey bars represent standard deviations. Pink backgrounds indicate significant differences between Pre- and post-RPT work tasks.

**Table 1 sensors-22-08008-t001:** T and *p* values and Cohen’s d effect sizes for each comparison performed between Task Initiation and Termination and pre- and post-RPT work tasks.

			Head	Sternum	Pelvis	Shoulder	Arm	Forearm	Hand
RPT	Acc.	0.4–0.6 Hz	t_23_ = −4.139 *p* < 0.001 Coh. d = 0.77	t_23_ = −2.955 *p* = 0.008 Coh. d = 0.69	t_23_ = −2.760 *p* = 0.011 Coh. d = 0.67	t_23_ = −1.215 *p* = 0.237 Coh. d = 0.21	t_23_ = −0.242 *p* = 0.811 Coh. d = 0.08	t_23_ = 0.678 *p* = 0.505 Coh. d = −0.08	t_23_ = 2.419 *p* = 0.024 Coh. d = −0.22
		6–12 Hz	t_23_ = −3.656 *p* = 0.001 Coh. d = 0.69	t_23_ = −2.370 *p* = 0.027 Coh. d = 0.34	t_23_ = −3.992 *p* < 0.001 Coh. d = 0.70	t_23_ = −1.442 *p* = 0.163 Coh. d = 0.15	t_23_ = −3.994 *p* < 0.001 Coh. d = 0.42	t23 = −1.141 *p* = 0.266 Coh. d = 0.20	t_23_ = −1.409 *p* = 0.172 Coh. d = 0.18
	Angular velocity	0.4–0.6 Hz	t_23_ = −6.103 *p* < 0.001 Coh. d = 0.98	t_23_ = −4.637 *p* < 0.001 Coh. d = 1.01	t_23_ = −3.499 *p* = 0.002 Coh. d = 0.78	t_23_ = −4.533 *p* < 0.001 Coh. d = 0.60	t_23_ = −1.002 *p* = 0.327 Coh. d = 0.18	t_23_ = 3.370 *p* = 0.003 Coh. d = −0.43	t_23_ = 2.366 *p* = 0.027 Coh. d = −0.32
		6–12 Hz	t_23_ = −3.754 *p* = 0.001 Coh. d = 0.79	t_23_ = −2.211 *p* = 0.037 Coh. d = 0.29	t_23_ = −3.866 *p* < 0.001 Coh. d = 0.58	t_23_ = −0.308 *p* = 0.761 Coh. d = 0.03	t_23_ = −2.862 *p* = 0.009 Coh. d = 0.31	t_23_ = −1.606 *p* = 0.122 Coh. d = 0.34	t_23_ = −2.323 *p* = 0.029 Coh. d = 0.38
Work task	Acc.	0.1–4 Hz	t_23_ = −2.526 *p* = 0.019 Coh. d = 0.35	t_23_ = −2.474 *p* = 0.021 Coh. d = 0.26	t_23_ = −2.121 *p* = 0.045 Coh. d = 0.28	t_23_ = −2.005 *p* = 0.057 Coh. d = 0.18	t_23_ = −2.256 *p* = 0.034 Coh. d = 0.18	t_23_ = −1.350 *p* = 0.190 Coh. d = 0.12	t_23_ = −1.726 *p* = 0.098 Coh. d = 0.15
		6–12 Hz	t_23_ = 0.637 *p* = 0.531 Coh. d = −0.04	t_23_ = 0.052 *p* = 0.959 Coh. d = 0.00	t_23_ = 0.409 *p* = 0.686 Coh. d = −0.02	t_23_ = 0.503 *p* = 0.620 Coh. d = −0.02	t_23_ = −0.164 *p* = 0.872 Coh. d = 0.01	t_23_ = −1.311 *p* = 0.203 Coh. d = 0.09	t_23_ = −1.168 *p* = 0.255 Coh. d = 0.08
	Angular velocity	0.1–4 Hz	t_23_ = −2.419 *p* = 0.024 Coh. d = 0.26	t_23_ = −2.541 *p* = 0.018 Coh. d = 0.33	t_23_ = −3.102 *p* = 0.005 Coh. d = 0.35	t_23_ = −2.011 *p* = 0.056 Coh. d = 0.20	t_23_ = −2.493 *p* = 0.020 Coh. d = 0.22	t_23_ = −1.399 *p* = 0.175 Coh. d = 0.12	t_23_ = −1.555 *p* = 0.134 Coh. d = 0.15
		6–12 Hz	t_23_ = 0.004 *p* = 0.999 Coh. d = 0.00	t_23_ = −0.603 *p* = 0.552 Coh. d = 0.05	t_23_ = −0.681 *p* = 0.503 Coh. d = 0.06	t_23_ = 0.343 *p* = 0.735 Coh. d = −0.02	t_23_ = −0.997 *p* = 0.329 Coh. d = 0.07	t_23_ = −1.690 *p* = 0.104 Coh. d = 0.13	t_23_ = −2.433 *p* = 0.023 Coh. d = 0.17

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
