# Peer review of "Power Spectrum of Acceleration and Angular Velocity Signals as Indicators of Muscle Fatigue during Upper Limb Low-Load Repetitive Tasks"

_sensors, 2022, doi:10.3390/s22208008_

Round 1
Reviewer 1 Report
This manuscript confirms the feasibility of using the frequency content of acceleration and angular velocity to evaluate muscle fatigue during low-load dynamic upper limb tasks. It is a topic of interest to the researchers in the related areas but the paper needs very significant improvement before acceptance for publication. My detailed comments are as follows:
1. Lack of specification introduction and picture display of Instrumentations, including the internal structure of IMU.
2. Algorithms and formulas can help explain the data processing process of power spectrum analysis.
3. Please recheck the pictures corresponding to the section of Maximum voluntary isometric force
4. There is a lack of detailed data analysis of the results to prove the effectiveness of acceleration and angular velocity in the evaluation of muscle fatigue
Author Response
Responses are provided in the attached document.

Reviewer 2 Report
In general, the paper is well written. The frequency-domain features are essential for muscle fatigue analysis. Several minor problems should be tackled before the final acceptance.
1. There are many low-load repetitive work tasks. Why did this work focus on tea bag filling?
2. Please provide the ROC curve, sensitivity, precision, and cut-off points for the analysis results. It could help future research for a better comparison.
Author Response
The responses are provided in the attached document.

Reviewer 3 Report
The review report for the manuscript “Power spectrum of acceleration and angular velocity signals as indicators of muscle fatigue during upper limb low-load repetitive tasks”
This work assesses the effect of muscle fatigue on the frequency content of acceleration and angular velocity signals during a standardized repetitive pointing task. Although the authors have done a lot of interesting work, there are still some problems that need to be improved.
1) How to understand the innovation of this paper? Please elaborate further.
2) The normality of the manuscript needs to be improved.
3) This article does not have a clear conclusion, giving the impression that the work is not done. It is suggested to modify the Abstract and the Conclusion parts.
Author Response

(The authors gave the same response as above.)

Round 2
Reviewer 1 Report
This is an significant study on power spectrum based on inertial measurement devices to assess muscle fatigue.The paper is well organized with clear figures and supporting charts. It is well written and the content is comprehensive. Overall, I agree to publish this work in Sensors.